# YAP1 and TAZ negatively control bone angiogenesis by limiting hypoxia-inducible factor signaling in endothelial cells

Kishor K Sivaraj[1,2], Backialakshmi Dharmalingam[1,2], Vishal Mohanakrishnan[1,2], Hyun-Woo Jeong[1,2], Katsuhiro Kato[1,2], Silke Schröder[1,2], Susanne Adams[1,2], Gou Young Koh[3,4,5], Ralf H Adams[1,2]*

[1]Department of Tissue Morphogenesis, Max Planck Institute for Molecular Biomedicine, Münster, Germany; [2]Faculty of Medicine, University of Münster, Münster, Germany; [3]Center for Vascular Research, Institute of Basic Science (IBS), Daejeon, Republic of Korea; [4]Graduate School of Medical Science and Engineering, Korea Advanced Institute of Science and Technology (KAIST), Daejeon, Republic of Korea; [5]Department of Biological Sciences, Korea Advanced Institute of Science and Technology (KAIST), Daejeon, Republic of Korea

**Abstract** Blood vessels are integrated into different organ environments with distinct properties and physiology (*Augustin and Koh, 2017*). A striking example of organ-specific specialization is the bone vasculature where certain molecular signals yield the opposite effect as in other tissues (*Glomski et al., 2011*; *Kusumbe et al., 2014*; *Ramasamy et al., 2014*). Here, we show that the transcriptional coregulators Yap1 and Taz, components of the Hippo pathway, suppress vascular growth in the hypoxic microenvironment of bone, in contrast to their pro-angiogenic role in other organs. Likewise, the kinase Lats2, which limits Yap1/Taz activity, is essential for bone angiogenesis but dispensable in organs with lower levels of hypoxia. With mouse genetics, RNA sequencing, biochemistry, and cell culture experiments, we show that Yap1/Taz constrain hypoxia-inducible factor 1α (HIF1α) target gene expression in vivo and in vitro. We propose that crosstalk between Yap1/Taz and HIF1α controls angiogenesis depending on the level of tissue hypoxia, resulting in organ-specific biological responses.

*For correspondence:
ralf.adams@mpi-muenster.mpg.
de

## Introduction

The skeletal system is surprisingly dynamic and undergoes lifelong remodeling even after the completion of developmental growth. The local vasculature plays central roles in embryonic and postnatal bone development but also later in homeostasis and in repair processes (*Gerber and Ferrara, 2000*; *Maes and Clemens, 2014*; *Zelzer and Olsen, 2005*). Endochondral ossification and, in particular, the formation of ossification centers in the mouse embryo requires the ingrowth of blood vessels into aggregates of hypertrophic chondrocytes that express pro-angiogenic signals such as vascular endothelial growth factor A (VEGF-A) (*Eshkar-Oren et al., 2009*; *Maes et al., 2010*). The invasion of blood vessels also facilitates the entry of osteoblast precursors into fractured bone and, accordingly, angiogenesis is essential for bone repair and regeneration (*Maes et al., 2010*; *Stegen et al., 2015*). It was also shown that capillaries in bone are regionally and molecularly specialized. With the help of advanced bone sample processing, staining and imaging protocols (*Kusumbe et al., 2015*), we found initially two distinct subpopulations of bone endothelial cells (ECs) that can be distinguished with the cell surface molecules CD31/Pecam1 and endomucin (Emcn) (*Kusumbe et al., 2014*). Columnar vessels in the metaphysis and the capillaries of the endosteum, a connective tissue layer lining the inner surface of compact bone, are associated with osteoprogenitor

cells expressing the transcription factor Osterix, express high levels of CD31 and Emcn (CD31[hi] Emcn[hi]), and have therefore been named type H. In contrast, the highly branched sinusoidal capillary network of the diaphysis shows comparably lower expression of the two markers (CD31[lo] Emcn[lo] or type L) and is predominantly associated with reticular mesenchymal cells and hematopoietic cells (*Kusumbe et al., 2014*; *Ramasamy et al., 2016b*). A further EC subpopulation with high CD31 expression – termed type E – is abundant in embryonic and early postnatal long bones but gradually disappears in adolescent mice (*Langen et al., 2017*). Indicating that they are source of signals controlling the behavior of perivascular mesenchymal cells, type H and type E but not type L ECs shift the differentiation of co-cultured primitive mesenchymal cells towards the osteoblast lineage (*Langen et al., 2017*). Interestingly, CD31[hi] Emcn[hi] ECs are strongly reduced in ovariectomized mice, a model of human postmenopausal osteoporosis, as well as in aging mice, which show an age-dependent decline in bone mineral density similar to human subjects (*Kusumbe et al., 2014*; *Xie et al., 2014*). Notch and hypoxia-inducible factor (HIF) signaling in ECs promote vessel growth in bone and help to maintain a functional type H EC population. Activation of these pathways in the ECs of aged mice induces the reappearance of CD31[hi] Emcn[hi] together with perivascular Osterix-positive cells and is sufficient to increase the formation of mineralized bone in the metaphysis (*Kusumbe et al., 2014*; *Ramasamy et al., 2016b*). Together, these reports indicate a strong coupling of EC behavior and osteogenesis in the skeletal system, which presumably involves a network of interdependent molecular interactions such as growth factors and matrix molecules.

Yes-associated protein 1 (Yap1) and WW domain containing transcription regulator 1 (WWTR1), which is more widely known as Taz, are closely related transcriptional coactivators and key components of the Hippo pathway, an evolutionarily conserved signaling cascade controlling cell proliferation, differentiation and organ size (*Dong et al., 2007*; *Huang et al., 2005*). In mammals, the core module of this pathway consists of the serine/threonine kinases Stk3 (Mst2) and Stk4 (Mst1), which are orthologues of *Drosphila* Hippo, the large tumor suppressor homolog 1/2 (Lats1/2) kinases, and their interaction partners Salvador (Sav1) and MOB kinase activator 1A/B (MOB1A/B) (*Piccolo et al., 2014*; *Yu and Guan, 2013*; *Zhao et al., 2011*). Activation of Hippo signaling leads to exclusion of Yap1/Taz from the nucleus and promotes the proteolytic degradation of these proteins. In the cell nucleus, Yap1/Taz interact with transcription factors, such as the TEA domain family members Tead1-4, and thereby regulate gene expression and promote growth processes (*Yu and Guan, 2013*). Accordingly, enhanced expression and nuclear localization of Yap1/Taz were observed in multiple human cancers (*Moroishi et al., 2015*) and inactivation of upstream Hippo signaling leads to tumor formation (*Lu et al., 2010*).

Here, we show that Yap1/Taz are critical regulators of vessel growth in an organ-dependent fashion. While inducible inactivation of the two genes in postnatal ECs results in the expected reduction of angiogenesis in the retinal vasculature, vessel growth and the abundance of CD31[hi] Emcn[hi] ECs are significantly increased in long bone. This also leads to an increase in metaphyseal bone formation, while EC-specific overexpression of a stabilized version of Yap1 or inactivation of the upstream kinase Lats2 impair angiogenesis. Furthermore, we show that Yap1/Taz negatively regulate the activity of the HIF pathway and thereby limit the expression of endothelial genes associated with vessel growth.

## Results

### Hypoxia and Yap1/Taz expression in bone endothelium

Different organs exhibit substantial variation in fundamental parameters such as tissue oxygenation (*Figure 1A,B*; *Figure 1—figure supplement 1A*), which is frequently enhanced in response to injury or in disease conditions (*De Santis and Singer, 2015*; *Samaja, 1988*). These differences are likely to reflect organ-specific features such as local oxygen consumption, blood flow, vascular architecture, and vessel diameter (*Figure 1—figure supplement 1B,C*). In comparison to other organs, such as brain, heart, lung, liver, spleen, or kidney, postnatal long bone contains extensive hypoxic areas. But even within long bone, pimonidazole (hypoxyprobe) staining or immunohistochemical analysis of the oxygen-controlled transcription factors hypoxia-inducible factor 1α (HIF-1α) and 2α (HIF-2α) uncover striking regional differences in oxygenation. The metaphyseal region in proximity of the growth plate is less hypoxic than the secondary ossification center (SOC) or the bone marrow in the diaphysis

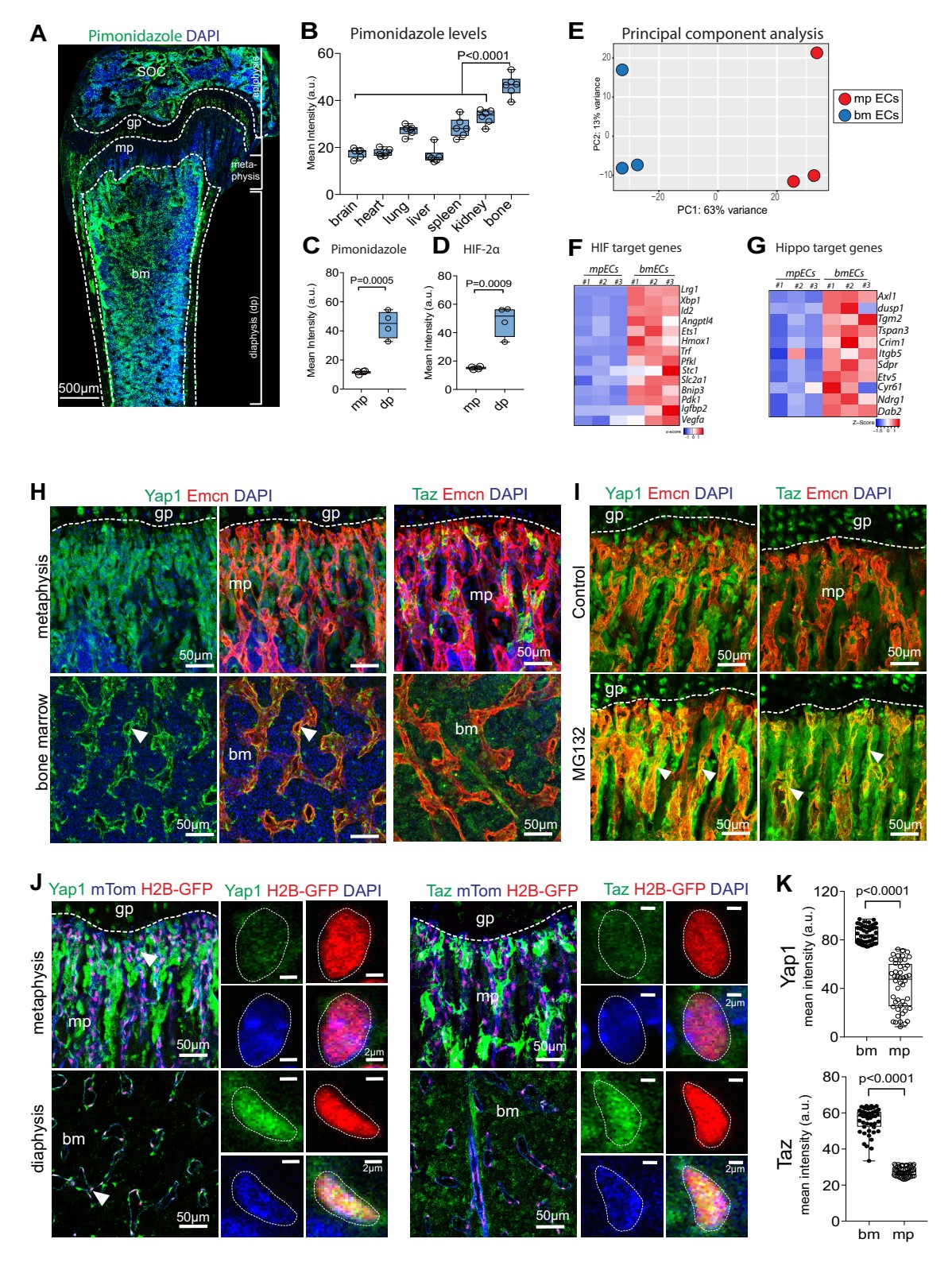

**Figure 1.** Regional differences in hypoxia and Yap1/Taz expression in bone. (**A**) Tile scan maximum intensity projection of P21 femur with Pimonidazole (green) and DAPI (blue) staining. (**B**) Quantification of Pimonidazole staining intensity (artificial units, a.u.) in different organs. (**C, D**) Regional differences in Pimonidazole (**C**) and HIF2α (**D**) staining levels in metaphysis (mp) and diaphysis (dp). (**E**) Principal component analysis of RNA sequencing data using most variable genes across the samples. The first principal component (PC1) explains 63% of all variance; and PC2 13% of the variance between

*Figure 1 continued on next page*

Figure 1 continued

metaphyseal (mpECs) and diaphyseal/bone marrow (bmECs) endothelial cells. (F, G) Heat map showing differential expression of hypoxia (F) and Yap1/Taz (G) controlled genes in mpECs vs. bmECs. (H) Confocal image showing immunostaining of Yap1 and Taz (H) in 3-week-old wild-type femur. Arrowheads highlight expression in Emcn+ (red) ECs. Nuclei, DAPI (blue). (I) Immunostaining of Yap1 and Taz in the control (vehicle) and MG132 proteasome inhibitor-treated femoral metaphysis. (J) Nuclear localization (arrowheads) of Yap1 (green) in H2B-GFP+ EC nuclei (shown in red) in 3-week-old *Cdh5-mTnG* femoral metaphysis (mp) and bone marrow (bm). Higher magnification image shows strong Yap1 and Taz nuclear signals bmECs. (K) Mean intensity (a.u.) of Yap1 and Taz nuclear localization signals in bm and mp ECs. (n = 4; 48 cells in total; data are presented as mean ±sem, P values, two-tailed unpaired *t-test*).

The online version of this article includes the following source data and figure supplement(s) for figure 1:

**Source data 1.** Source data for *Figure 1B,C,D,F,G,K*.
**Figure supplement 1.** Bone-specific features of the vasculature.
**Figure supplement 1—source data 1.** Source data for *Figure 1—figure supplement 1C*.
**Figure supplement 2.** Spatial molecular differences between the metaphyseal and BM vasculature.
**Figure supplement 2—source data 1.** Source data for *Figure 1—figure supplement 2E,F,G,H and I*.
**Figure supplement 3.** Hippo signaling in bone ECs.
**Figure supplement 3—source data 1.** Source data for *Figure 1—figure supplement 3E*.

(bone shaft) (*Figure 1A–D*; *Figure 1—figure supplement 1D,E* and *Figure 1—source data 1*). As the consequences of such regional differences are not understood, we isolated endothelial cells (ECs) expressing membrane-anchored mTomato fluorescent protein and nuclear H2B-GFP from *Cdh5-mT/nG* transgenic reporter femoral metaphysis and diaphyseal bone marrow cavity by fluorescence-activated cell sorting (FACS) at high purity for RNA sequencing (*Figure 1—figure supplement 1F,G*; *Figure 1—figure supplement 2A* and *Figure 1—source data 1*). Principal component analysis (PCA) of RNA-seq data showed separate clusters of metaphyseal ECs (mpECs) and bone marrow ECs (bmECs) with less variation in gene expression within each group (*Figure 1E*; *Figure 1—figure supplement 2B–D*). Endothelial marker genes such as *Pecam1, Emcn, Cd34, Aplnr, Efnb2*, and *Esm1* are upregulated in mpECs relative to bmECs, whereas *Kdr, Flt4, Tek, and Sele* are downregulated (*Figure 1—figure supplement 2E–G* and *Figure 1—figure supplement 1—source data 1*). Consistent with the immunostaining results (*Figure 1—figure supplement 1D,E*), transcripts for hypoxia-inducible transcription factors, namely *Hif1a* and *Epas1 (Hif2a)*, and oxygen-controlled target genes are upregulated in bmECs (*Figure 1F* and *Figure 1—figure supplement 2H*). In line with the regional differences in oxygenation, which are likely to limit the ability to metabolize fatty acids, genes belonging to the glycolytic pathway are higher in bmECs than in mpECs (*Figure 1—figure supplement 2I* and *Figure 1—figure supplement 1—source data 1*). Our analysis also uncovered that target genes of Hippo signaling, an evolutionarily conserved pathway that promotes cell proliferation and organ growth, are upregulated in bmECs (*Figure 1G*). In sections of P21 femur, immunostaining of Yes-associated protein 1 (Yap1), a transcriptional coregulatory in the Hippo signaling cascade, is more prominently visible in BM sinusoidal ECs, whereas expression in the metaphysis predominates in perivascular cells (*Figure 1H*; *Figure 1—figure supplement 3A*). Immunosignals for the Yap1-related protein Taz mainly decorate non-endothelial cells in the metaphysis and arterial ECs (*Figure 1H*; *Figure 1—figure supplement 3B*). Given that proteolytic degradation limits the biological activity of Yap1/Taz (*Piccolo et al., 2014*; *Yu and Guan, 2013*; *Zhao et al., 2011*), we examined whether acute inhibition of proteasome-dependent degradation would highlight sites where Yap1/Taz levels are actively suppressed. Treatment of 3-week-old mice with the proteasome inhibitor MG132 for 3 hr profoundly increased endothelial and perivascular Yap1 and Taz protein signals in the metaphysis but had relatively limited effects in the diaphysis (*Figure 1I*; *Figure 1—figure supplement 3C*). Further arguing for rapid degradation of Yap1/Taz in the metaphysis, strong phospho-Yap1$^{S127}$ immunosignals can be seen in metaphyseal capillaries and associated perivascular cells, which is accompanied by high expression of Lats2 protein in metaphyseal but not diaphyseal vessels (*Figure 1—figure supplement 3D*). In contrast, *Yap1, Wwtr1, Lats1*, and *Lats2* transcripts are not significantly different between EC subsets (*Figure 1—figure supplement 3E*). In *Cdh5-mT/nG* reporter mice, Yap1 and Taz immunostaining is prominent in the H2B-GFP+ nuclei of bmECs or in cultured bone ECs in vitro, whereas comparably little signal can be seen in mpECs (*Figure 1J,K*; *Figure 1—figure supplement 3F*). Together, these data indicate that Hippo signaling is active in ECs of the metaphysis leading to rapid degradation of Yap1/Taz in these cells.

## Function of Yap1/Taz in organ-specific angiogenesis

Next, we used tamoxifen inducible *Cdh5-Cre-ERT2* mice, which allow efficient genetic experiments in all bone ECs in vivo (*Kusumbe et al., 2014*; *Langen et al., 2017*), to generate EC-specific Yap1 and Taz double loss-of-function mutants (Yap1/Taz$^{i\Delta EC}$). Following administration of tamoxifen from postnatal day (P) 6–8, Yap1/Taz$^{i\Delta EC}$ long bones at P21 display a much denser vessel network with more endothelial buds and column, a feature of actively growing bone vessels (*Ramasamy et al., 2016a*), relative to littermate controls (*Figure 2A,B*; *Figure 2—figure supplement 1A*). Yap1/Taz$^{i\Delta EC}$ ECs show reduced expression of both *Yap1* and *Wwtr1* at the transcript levels and endothelial Yap1 is also reduced by immunostaining (*Figure 2—figure supplement 1A–C*). A second administration regime, in which tamoxifen is given from P1-3, is incompatible with survival of Yap1/Taz$^{i\Delta EC}$ animals until P21 and therefore these mutants were analyzed at P18. This showed that the density of Yap1/Taz$^{i\Delta EC}$ bone vessels is strongly increased at P18, which is particularly evident at the interface (transition zone) between the metaphyseal and diaphyseal capillary networks (*Langen et al., 2017*), and sinusoidal marrow vessels are enlarged (*Figure 2—figure supplement 1D–F*). Furthermore, the number of total ECs, identified by Emcn immunostaining in combination with the *Cdh5-mT/nG* reporter, of Emcn+ EdU+ proliferating ECs, and of αSMA-covered arteries are significantly increased in P21 Yap1/Taz$^{i\Delta EC}$ long bone relative to littermate controls (*Figure 2C,D*; *Figure 2—figure supplement 1G,H*). The number of vessel buds and column in proximity of the growth plate and the length of columnar vessels, all of which are associated with vessel growth (*Kusumbe et al., 2014*; *Ramasamy et al., 2016b*), are also increased in Yap1/Taz$^{i\Delta EC}$ long bone (*Figure 2A,B,E,F*). The analysis of EC subpopulations by flow cytometry (*Figure 2—figure supplement 2D*) and immunostaining revealed that the increase is more pronounced for Yap1/Taz$^{i\Delta EC}$ total and type H ECs, but also the fraction of type L ECs is significantly larger than in littermate controls (*Figure 2E*; *Figure 2—figure supplement 1I*).

Further arguing that Yap1/Taz play an unusual role as negative regulators of bone angiogenesis, *Cdh5-CreERT2*–controlled and thereby inducible overexpression of a stabilized version of Yap1 (Yap1$^{S112A}$) in ECs (Yap1-KI$^{iEC}$) (*Figure 2—figure supplement 2A–C*) impairs growth of the bone vasculature. In P21 Yap1-KI$^{iEC}$ femur, distal vessel buds and arches, metaphyseal type H vessel columns, and EC proliferation are reduced (*Figure 2G–L*). To study the upstream regulation of Yap1/Taz activity in bone ECs, we generated tamoxifen-inducible EC-specific Lats2$^{i\Delta EC}$ mutants by combining *Cdh5-Cre-ERT2* transgenic mice and animals carrying loxP-flanked *Lats2* alleles (*Lu et al., 2010*). The absence of *Lats2* in ECs causes a striking reduction of P21 femur and tibia length and weight (*Figure 3A–C*; *Figure 3—figure supplement 1A–C*). Growth of the Lats2$^{i\Delta EC}$ long bone vasculature and vascularization of the secondary ossification center are strongly compromised, and vessel columns, distal buds as well as total type H ECs are significantly reduced compared to littermate controls (*Figure 3D,E*; *Figure 3—figure supplement 1D–G*). In line with the role of Lats2 as a negative regulator of Yap1 and Taz, both proteins are increased in Lats2$^{i\Delta EC}$ metaphyseal vessels (*Figure 3F,G*; *Figure 3—figure supplement 1H–J*). Vessel buds mediating angiogenic growth are lost or converted in mutant long bone so that they resemble vessel sprouts seen in other organs (*Figure 3F,H*; *Figure 3—figure supplement 1J,K*). In line with the reduction of Lats2$^{i\Delta EC}$ bone vessels, mpEC proliferation is significantly reduced, whereas apoptotic ECs are increased (*Figure 3I–M*).

The murine retina is a widely used model of postnatal sprouting angiogenesis, in which the growth of an initially two-dimensional superficial vessel plexus can be easily monitored from P1 to P7 (*Gerhardt et al., 2003*; *Pitulescu et al., 2010*). Immunostaining of Yap1 and Taz in P6 retina confirmed expression of both proteins in the vascular endothelium (*Figure 3—figure supplement 2A, B*). Following the administration of tamoxifen from P1-P3, the outgrowth of the retinal vasculature, vessel sprouting and branching are impaired in Yap1/Taz$^{i\Delta EC}$ double mutant retinas, which show strongly reduced EC numbers and significantly fewer proliferating (EdU+ Erg+) ECs (*Figure 3—figure supplement 2C–E*). These findings are consistent with recently published results (*Kim et al., 2017*; *Neto et al., 2018*) and show that Yap1 and Taz have opposite roles in the regulation of vessel growth in retina and bone. Lats2$^{i\Delta EC}$ mutants show strong defects in the bone vasculature but not in other organs such as retina, brain, liver, lung, kidney, heart, and spleen (*Figure 3—figure supplement 3A–D*). These data establish that endothelial Yap1/Taz and Lats2 have distinct roles in different organs.

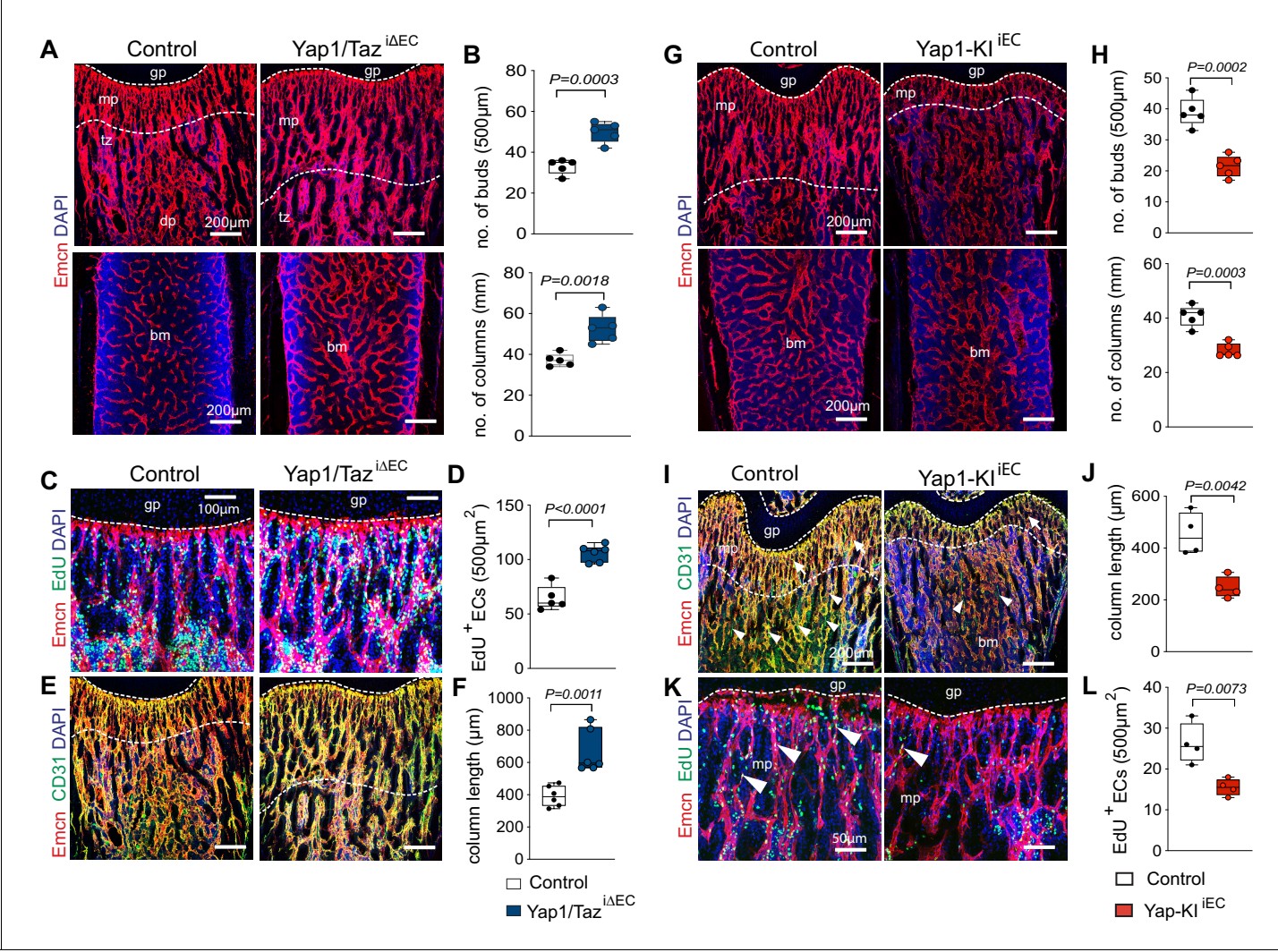

**Figure 2.** Yap1/Taz inhibits angiogenesis in bone. (**A, B**) Representative confocal images of P21 control and Yap1/Taz[iΔEC] femoral Emcn+ (red) vasculature (**A**). Nuclei, DAPI (blue). Metaphysis (mp), transition zone (tz), diaphysis (dp), and growth plate (gp) are indicated. Note the increased number of Yap1/Taz[iΔEC] vessel buds and columns (**B**) compared to littermate control (n = 6, data are presented as mean ±sem. *P* values, two-tailed unpaired *t-test*). (**C, D**) Representative confocal image of Emcn+ (red) proliferating (EdU, green) mpECs. Nuclei, DAPI (blue) (**C**). Quantification of EdU+ Emcn+ ECs in Yap1/Taz[iΔEC] and control metaphysis (**D**), (control n = 6 and Yap1/Taz[iΔEC] n = 7, data are presented as mean ±sem. *P* values, two-tailed unpaired *t-test*). (**E, F**) Maximum intensity projections of Emcn[hi] (red) CD31[hi] (green) vessels in the P21 Yap1/Taz[iΔEC] and control femur (**E**). Metaphyseal column length is significantly increased in Yap1/Taz[iΔEC] mutant compared to control femur (**F**) (n = 6 data are presented as mean ±sem. *P* values, two-tailed unpaired *t-test*). (**G, H**) Representative confocal images of control and Yap1-KI[iEC] femur. Emcn+ (red) ECs and nuclei (DAPI, blue) are stained (**G**). Vessel buds and columns are reduced in the Yap1-KI[iEC] metaphysis relative to littermate control (**H**) (n = 5, data are presented as mean ±sem. *P* values, two-tailed unpaired *t-test*). (**I, J**) Maximum intensity projection of Emcn[hi] (red) CD31[hi] (green) vessels in P21 Yap1-KI[iEC] and control femur. The vasculature of the metaphysis (mp) (arrows; dashed lines), the transition zone (tz) connecting the mp to the diaphysis (dp) and arteries (arrowheads) are reduced in *Yap1* gain-of-function femur (**I**). The length of the Yap1-KI[iEC] Emcn[hi] CD31[hi] vessel columns in femur is significantly reduced (**J**) (control n = 4 data are presented as mean ±sem. *P* values, two-tailed unpaired *t-test*). (**K, L**) Representative confocal image of proliferating ECs (Emcn, red; EdU, green) in femoral metaphysis. Nuclei, DAPI (blue) (**K**). Quantification of EdU+ Emcn+ ECs in Yap1-KI[iEC] and control metaphysis (**L**) (control n = 4 data are presented as mean ±sem. *P* values, two-tailed unpaired *t-test*).

The online version of this article includes the following source data and figure supplement(s) for figure 2:

**Source data 1.** Source data for *Figure 2B,D,F,H,J,L*.
**Figure supplement 1.** Endothelial Yap1/Taz in bone angiogenesis.
**Figure supplement 1—source data 1.** Source data for *Figure 2—figure supplement 1B,E,G,H,I*.
**Figure supplement 2.** Overexpression of Yap1 in ECs inhibits angiogenesis in bone.

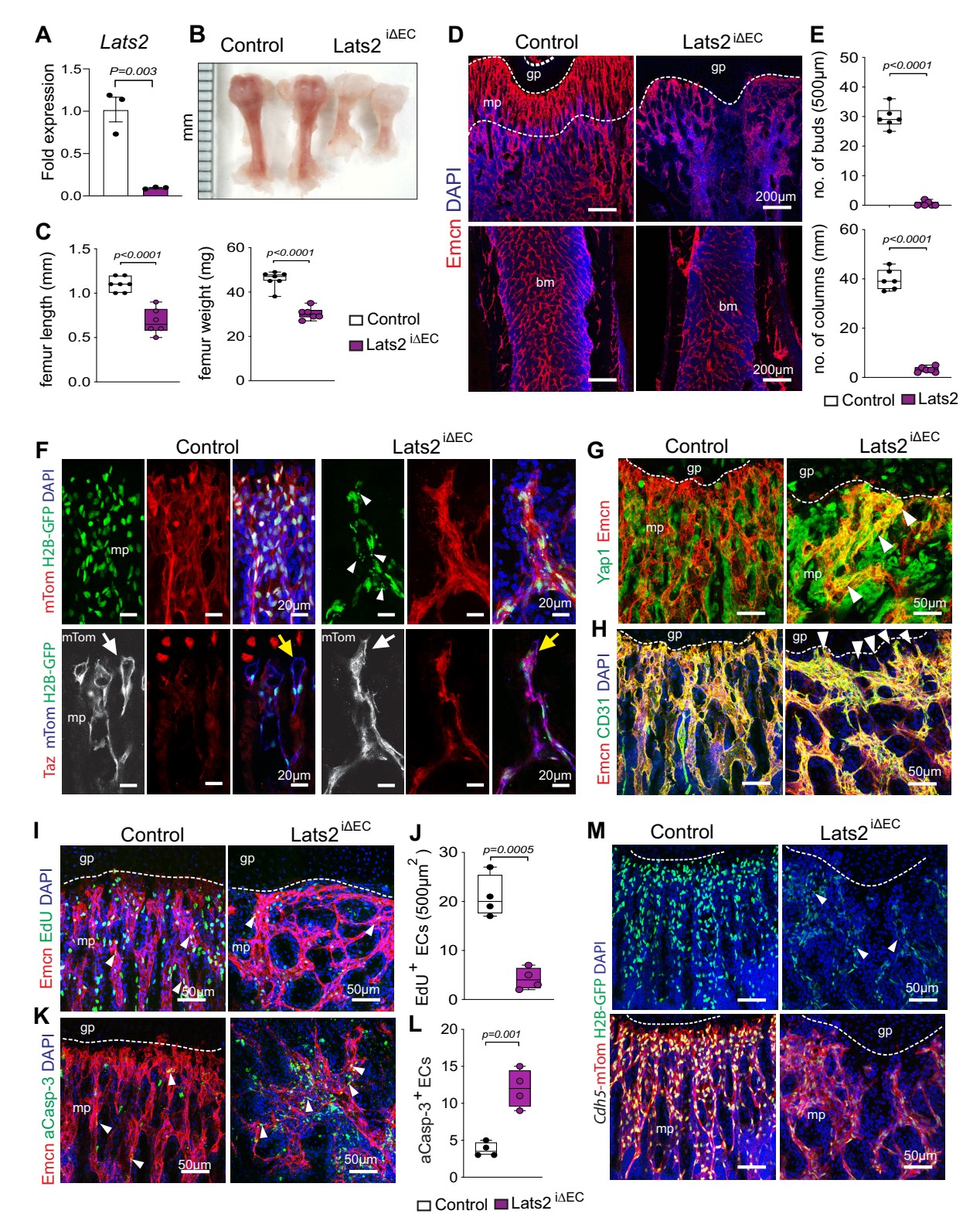

**Figure 3.** Endothelial Lats2 promotes angiogenesis in bone. (**A**) *Lats2* transcript levels are significantly decreased in freshly isolated Lats2$^{i\Delta EC}$ mutant bone ECs compared to control. (n = 3, data are presented as mean ±sem. *P* values, two-tailed unpaired *t-test*). (**B, C**) Freshly dissected P21 Lats2$^{i\Delta EC}$ mutant femur relative to littermate control (**B**). Femur length (mm) and weight are reduced in Lats2$^{i\Delta EC}$ mutants (**C**). (control n = 7 and mutant n = 6; data are presented as mean ±sem, *P* values, two-tailed unpaired *t-test*). (**D, E**) Representative confocal images of P21 control and Lats2$^{i\Delta EC}$ femur

*Figure 3 continued on next page*

Figure 3 continued

stained with Emcn+ (red) vasculature, and nuclei in DAPI (blue) (D). Vessel buds and columns are strongly reduced in Lats2$^{i\Delta EC}$ mutants compared to littermate controls (E). (n = 6, data are presented as mean ±sem. *P* values, two-tailed unpaired *t-test*). (F) Lats2$^{i\Delta EC}$ distal vessel buds and arches switch to a tip-like morphology (arrow). The *Cdh5-mTnG* reporter (red and green) visualizes nuclear fragmentation (arrowheads) in Lats2$^{i\Delta EC}$ ECs but not in control. Taz immunosignal is strongly increased in Lats2$^{i\Delta EC}$ mutants. (G) Yap1 (green) immunosignal is enhanced in Lats2$^{i\Delta EC}$ metaphyseal vessels (Emcn, red). (H) Confocal image of control and Lats2$^{i\Delta EC}$ femoral metaphysis stained for Emcn (red), CD31 (green), and nuclei (DAPI, blue). (I, J) Representative confocal image of proliferating (Emcn+, red; EdU+, green) ECs in metaphysis. Nuclei (DAPI, blue) (I). Quantification of EdU$^+$ Emcn$^+$ ECs in Lats2$^{i\Delta EC}$ and control metaphysis (J) (control n = 4 and Lats2$^{i\Delta EC}$n = 4 data are presented as mean ±sem. *P* values, two-tailed unpaired *t-test*). (K, L) Apoptotic Emcn+ (red) and active caspase-3+ (aCasp-3, green) ECs in metaphysis. Nuclei (DAPI, blue) (K). Quantification of Emcn$^+$ aCasp-3$^+$ ECs (L) (control n = 4 and *Lats2$^{\Delta EC}$*n = 4 data are presented as mean ±sem. *P* values, two-tailed unpaired *t-test*). (M) Representative confocal image of Lats2$^{i\Delta EC}$ *Cdh5-mTnG* femur compared to littermate control.

The online version of this article includes the following source data and figure supplement(s) for figure 3:

**Source data 1.** Source data for *Figure 3A,C,E,J,L*.
**Figure supplement 1.** Endothelial Lats2 promotes bone angiogenesis.
**Figure supplement 1—source data 1.** Source data for *Figure 3—figure supplement 1B,C,G*.
**Figure supplement 2.** Endothelial Yap1/Taz in retinal angiogenesis.
**Figure supplement 2—source data 1.** Source data for *Figure 3—figure supplement 2E*.
**Figure supplement 3.** Endothelial Lats2 in organ-specific angiogenesis.

## Crosstalk between Yap1/Taz and HIF-1α function

To gain insight into the surprising increase in bone angiogenesis in the absence of Yap1/Taz, we performed RNA sequencing (RNA-seq) analysis of freshly isolated bone ECs from P21 Yap1/Taz$^{i\Delta EC}$ and littermate controls in triplicates. Principal component analysis (PCA) of the resulting RNA-seq data shows 62% variation between control and mutant samples, which cluster into two distinct groups (*Figure 4—figure supplement 1A*). Differential gene expression adjusted to a p-value of 0.05 identifies 341 upregulated genes and 118 downregulated genes in Yap1/Taz$^{i\Delta EC}$ mutant ECs (*Figure 4A*; *Figure 4—figure supplement 1B* and *Figure 4—figure supplement 1—source data 1*). Gene ontology (GO) analysis shows biological processes such as angiogenesis, vascular development and EC migration as top hits among the genes that are upregulated in Yap1/Taz$^{i\Delta EC}$ ECs (*Figure 4—figure supplement 1C*). Gene set enrichment analysis (GSEA) for cellular signaling pathways reveals that the hypoxia-inducible factor (HIF1α and HIF2α) transcription factor signaling networks are enriched (*Figure 4B*). In particular, HIF1α target genes and HIF-controlled pro-angiogenic genes, such as *Vegfa*, *Lrg1*, *Angtpl4*, and *Xbp1*, are strongly upregulated in Yap1/Taz$^{i\Delta EC}$ bone ECs (*Figure 4C,D* and *Figure 4—source data 1*). Bone ECs from Lats2$^{i\Delta EC}$ mutants, which show a profound reduction of the bone vasculature, were also analyzed by RNA-seq (*Figure 4—figure supplement 1D,E*). Consistent with the mutant phenotypes, expression of the Yap1/Taz controlled genes *Cyr61*, *Thbs1*, *Ddah1*, *Lhfp*, and *Ctrg* is increased in Lats2$^{i\Delta EC}$ ECs, whereas the HIF1α controlled genes *Lrg1*, *Xbp1*, *Angptl4*, and *Vegfa* are downregulated (*Figure 4E–G*). Genes characteristic for bmECs are strongly downregulated (*Figure 4—figure supplement 1F*), but markers of tip cells in EC sprouts and as well pro-apoptotic genes are up-regulated in Lats2$^{i\Delta EC}$ ECs (*Figure 4—figure supplement 1F–H* and *Figure 4—figure supplement 1—source data 1*). Consistent with the higher levels of Yap1/Taz in bmECs (*Figure 1J,K*), the loss of *Ctgf* and *Cyr61* expression and the upregulation of *Vegfa* and *Angptl4* are more pronounced in freshly sorted Yap1/Taz$^{i\Delta EC}$ type L ECs relative to type H ECs. Conversely, *Ctgf* and *Cyr61* transcripts are more strongly increased and *Vegfa* and *Angptl4* reduced in Lats2$^{i\Delta EC}$ type L ECs relative to type H ECs (*Figure 4H,I*).

To understand the underlying mechanism for the surprising function of Yap1/Taz in bone ECs, we used an independent approach in cultured human umbilical vein endothelial cells (HUVEC). HUVECs grown under hypoxic (1% O$_2$) conditions strongly upregulate *VEGFA* and *ANGPTL4* expression relative to normoxia (21% O$_2$). Hypoxia also leads to strong upregulation of the Yap1/Taz target genes *CTGF* and *CYR61* (*Figure 5A*) and, consistently, Yap1/Taz nuclear localization is increased under these conditions (*Figure 5B*). In vitro siRNA-mediated knockdown (KD) of human *YAP1* and *TAZ* efficiently reduces their expression at the transcript and protein level (*Figure 5—figure supplement 1A,B*). Expression of the Hippo pathway target genes *CTGF* and *CYR61* is strongly reduced and the expression of the HIF1α target genes *VEGFA* and *XBP1* are increased in *siYAP1/TAZ* treated cells (*Figure 5C*; *Figure 5—figure supplement 1C*). Moreover, immunoprecipitation and Western blot

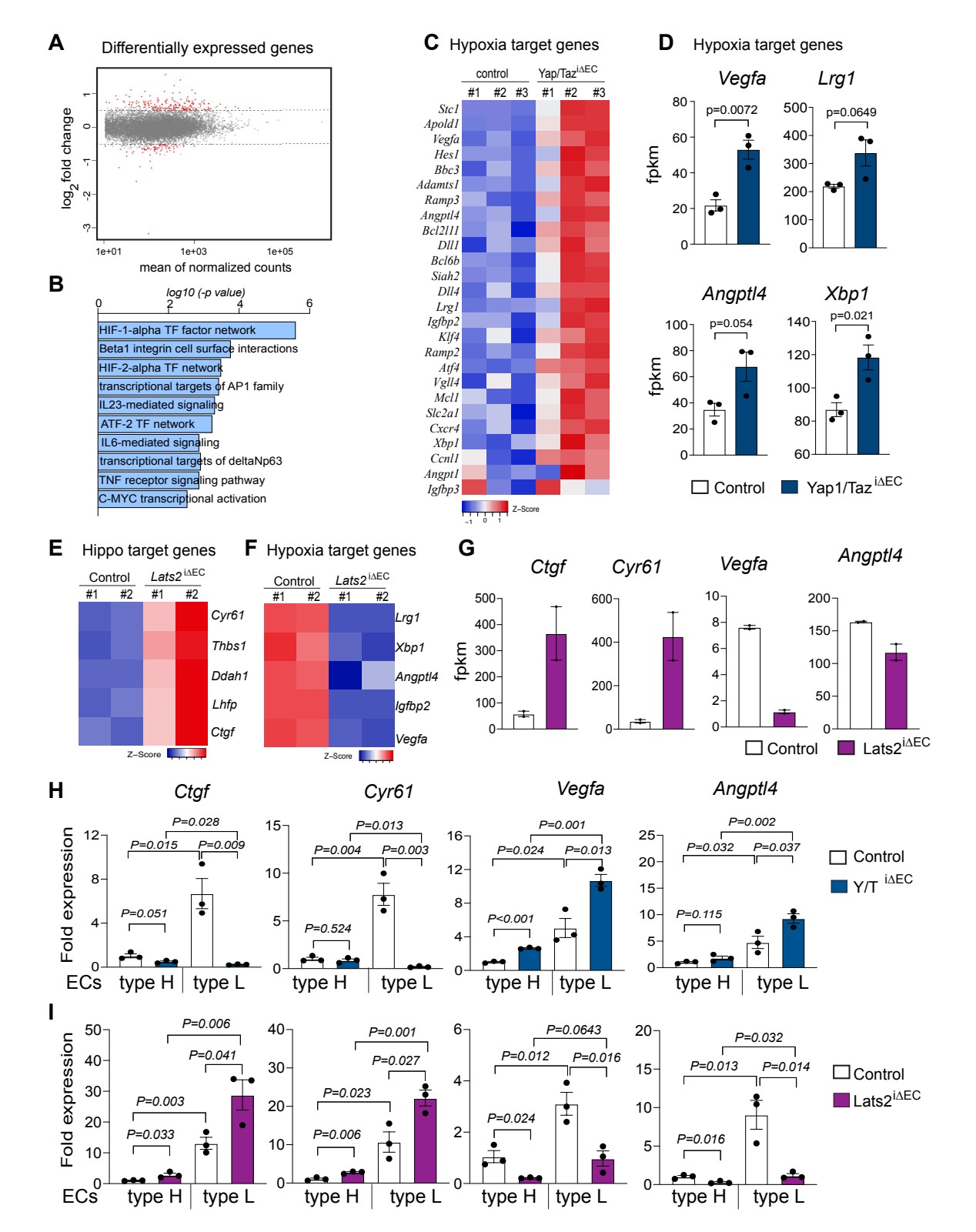

**Figure 4.** Loss of Yap1/Taz changes hypoxia target gene expression in bone ECs. (**A**) Differentially expressed genes determined by RNA-seq analysis of ECs from 3-week-old control vs. Yap1/Taz$^{i\Delta EC}$ mutant bone (n = 3; log2 fold change). (**B**) Bar graph showing enrichment of cellular signaling pathway components among Yap1/Taz$^{i\Delta EC}$ upregulated genes. (**C**) Heatmap of upregulated HIF1α target genes in Yap1/Taz$^{i\Delta EC}$ bone ECs. (**D**) RNA-seq data showing expression of the HIF1α target genes *Vegfa*, *Lrg1*, *Angptl4*, and *Xbp1* in control and mutant bone ECs (n = 3; data are presented as

*Figure 4 continued on next page*

Figure 4 continued

mean ±sem). (E, F) Heatmaps of Hippo pathway (E) and hypoxia (F) target genes. Expression of *Cyr61, Thbs1, Ddah1, Lhfp, and Ctgf* is increased in Lats2[iΔEC] ECs relative to control, whereas the HIF1α targets *Lrg1, Xbp1, Angptl4, Igfbp2, and Vegfa* are decreased. (G) Bar graph showing *Ctgf, Cyr61, Vegfa* and *Angptl4* gene expression levels in control and Lats2[iΔEC] mutant ECs. (H, I) Expression of the Yap1/Taz targets *Ctgf, Cyr61* and the HIF1α targets *Vegfa, Angptl4* in freshly isolated type H and type L EC subpopulations from P21 Yap1/Taz[iΔEC] (H) and Lats2[iΔEC] (I) bone samples.

The online version of this article includes the following source data and figure supplement(s) for figure 4:

**Source data 1.** Source data for *Figure 4C,D,E,F,G,H,I*.
**Figure supplement 1.** The Hippo pathway regulates HIF1α-controlled gene expression.
**Figure supplement 1—source data 1.** Source data for *Figure 4—figure supplement 1B,F,G,H*.

experiments show that HIF1α can be co-immunoprecipitated with Yap1/Taz proteins under hypoxic conditions (*Figure 5D*). To test whether the *siYAP/TAZ*-induced upregulation of *VEGFA* requires HIF1α, we used siRNA to interfere with *HIF1A* expression in HUVECs. Indeed, the HIF1α target genes *VEGFA, ANPTL4, IGFBP2* and *XBP1* are no longer upregulated after triple knockdown of *HIF1A* together with *YAP1/TAZ* (*Figure 5E*; *Figure 5—figure supplement 1C*). Moreover, ChIP experiments show that the binding of HIF1α to a hypoxia response element in the VEGFA gene (position −947) is significantly increased after *YAP1/TAZ* knockdown (*Figure 5F*). Highlighting the crosstalk between Yap1/Taz and HIF1α further, EC-specific inactivation of the *Hif1a* gene impairs the expression of *Vegfa* and *Angptl4* both in type H and type L ECs in vivo, but it also reduces the expression of *Ctgf* and *Cyr61* in type L cells (*Figure 5—figure supplement 1D*).

HIF is stabilized under hypoxic conditions, translocates to the nucleus and regulates gene expression to stimulate angiogenesis and other processes (*Pugh and Ratcliffe, 2003*). In contrast, HIF α subunits are modified by prolyl-hydroxylases and thereby marked for rapid proteolytic degradation in normoxia (*Aragonés et al., 2009*; *Safran and Kaelin, 2003*). *Cdh5-CreERT2*-controlled and thereby EC-specific expression of a stabilized HIF1α mutant (Hif1dPA[iEC]) lacking critical prolyl-hydroxylation sites (*Kim et al., 2006*) increases the abundance of metaphyseal type H capillaries, of columns and buds in proximity of the growth plate, and of arteries (*Figure 6A–C*), which is consistent with previously reported findings on the role of HIF signaling in bone ECs (*Kusumbe et al., 2014*). Supporting the crosstalk between Yap1/Taz and HIF-1α in vivo, the combination of Yap1/Taz[iΔEC] and Hif1a[iΔEC] loss-of-function alleles normalizes the increase in bone vascularization seen in Yap1/Taz[iΔEC] samples (*Figure 6D,E*).

## Endothelial Hippo signaling controls osteogenesis

Based on previous studies revealing that bone angiogenesis and osteogenesis are coupled, we next investigated whether the changes in the Yap1/Taz[iΔEC] femoral vasculature results in altered bone formation. P21 Yap1/Taz[iΔEC] mutant mice display slightly lower body weights than control littermates, whereas the length and weight of their femurs and tibias are not changed significantly (*Figure 7A–C*; *Figure 7—figure supplement 1A*). Osterix+ and Runx2+ osteoprogenitor cells and multiple indicators of bone formation, such as osteocalcin and the matrix proteins osteopontin and collagen type I, are increased in the Yap1/Taz[iΔEC] femur (*Figure 7D,E*; *Figure 7—figure supplement 1B*). Micro-computed tomography (μCT) analysis of Yap1/Taz[iΔEC] femurs demonstrated that these changes are accompanied by significant increases in trabecular bone (Tb) fraction volume, Tb number and thickness in comparison to littermate controls (*Figure 7H,I*). Trabecular spacing, a parameter reflecting the separation of Tb elements, is reduced in Yap1/Taz[iΔEC] samples. The number of osteoclasts, identified by antibodies directed against a subunit of vacuolar ATPase (ATP6V1B1/B2), is increased in mutants (*Figure 7—figure supplement 1B*) arguing that it is unlikely that the increase in trabecular bone is caused by impaired bone degradation.

Consistent with the reduced vascular growth after EC-specific expression of stabilized Yap1[S112A], bone formation is also reduced in Yap1-KI[iEC] femurs. Body weight as well as femoral and tibial length and weight is reduced in P21 Yap1-KI[iEC] mutants (*Figure 7—figure supplement 1C–E*). Expression of Yap1[S112A] also leads to the reduction of Osterix+ osteoprogenitor cells and of osteopontin immunostaining (*Figure 7F,G*). The increase in Osterix+ cells seen after the loss of endothelial Yap1/Taz is restored to control level by the simultaneous and EC-specific inactivation of *Hif1a* (*Figure 7—figure supplement 1F*).

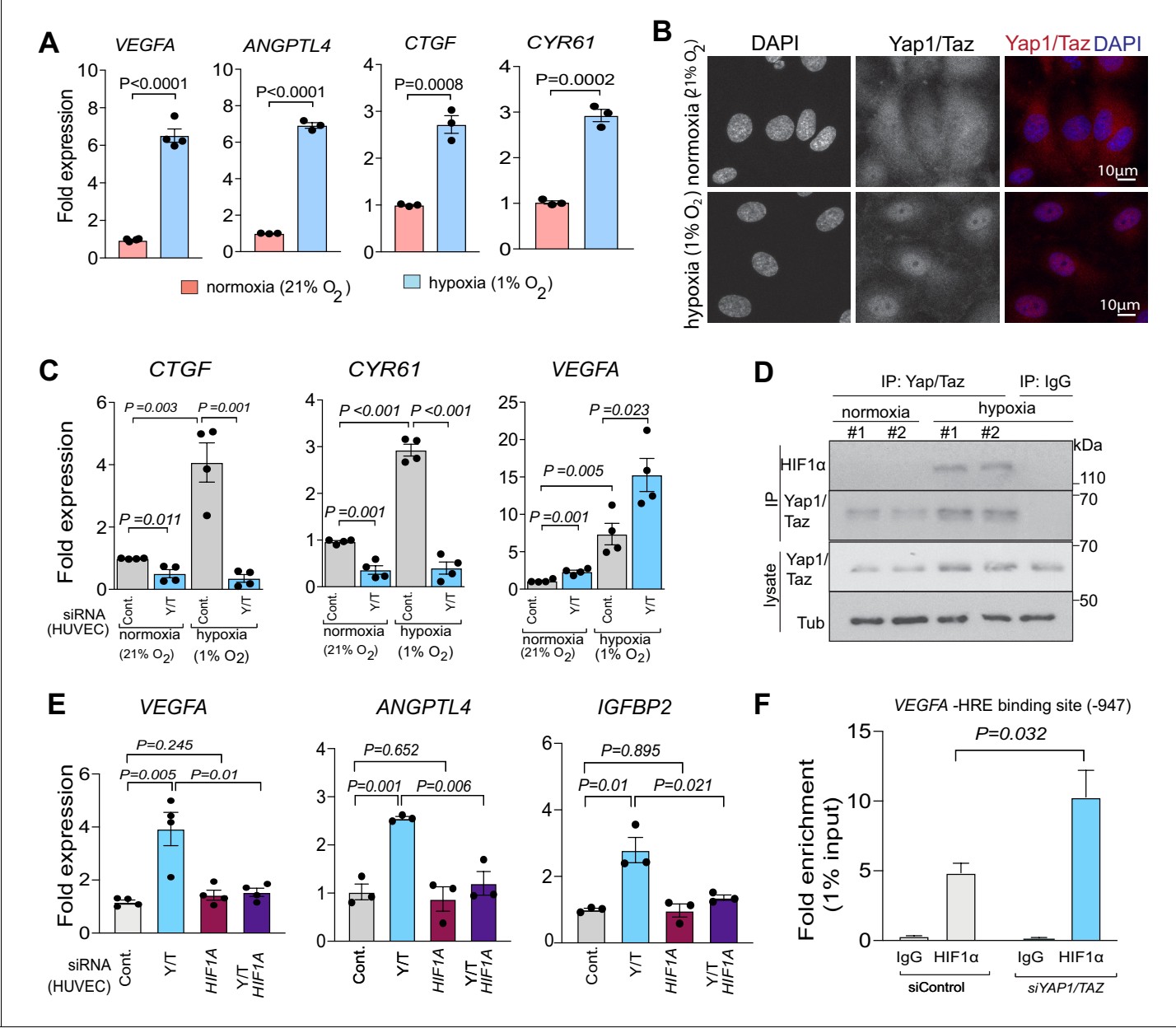

**Figure 5.** Yap1 and Taz inhibit HIF-1α-controlled gene expression. (**A**) Increased expression of the HIF1α target genes *VEGFA* and *ANGPTL4* as well as Yap1/Taz target genes *CTGF* and *CYR61* under hypoxic condition (1% O₂) compared to normoxia (21% O₂) (n = 4; data are presented as mean ±sem, *P* values, two-tailed unpaired *t*-test). (**B**) Confocal image of HUVEC showing nuclear accumulation of Yap1/Taz in 1% O₂ relative to 21% O₂. Nuclei, DAPI (blue). (**C**) Increased expression of the Yap1/Taz target genes *CTGF, CYR61* and the HIF1α target *VEGFA* in HUVECs in 1% O₂ relative to 21% O₂. *CTGF* and *CYR61* expression under both conditions is significantly reduced in *siYAP1/TAZ*-transfected (Y/T) HUVECs. *VEGFA* expression significantly increased in Y/T cells in both conditions (n = 4; data are presented as mean ±sem, *P* values, two-tailed unpaired *t*-test). (**D**) Western blots showing immunoprecipitation (IP) of Yap1/Taz followed by immunoblotting of HIF1α and Yap1/Taz. IgG is used as negative control in IP. Lysates are shown as loading control. (**E**) Increased *VEGFA, ANGPTL4, IGFBP2* expression in *siYAP1/TAZ*-treated HUVECs is normalized by *siHIF1A* transfection, whereas baseline *VEGFA, ANGPTL4* and *IGFBP2* is not altered by *HIF1A* knockdown alone (n = 3–4; data are presented as mean ±sem, *P* values, two-tailed unpaired *t*-test). (**F**) Enrichment of *VEGFA* promoter sequences after chromatin immunoprecipitation with HIF1α antibodies compared to IgG antibodies under hypoxic conditions and after transfection of HUVECs with *siYAP1/TAZ* or *siControl* (n = 3; data are presented as mean ±sem, *P* values, two-tailed unpaired *t*-test).

The online version of this article includes the following source data and figure supplement(s) for figure 5:

**Source data 1.** Source data for *Figure 5A,C,E,F*.

**Figure supplement 1.** Yap1 and Taz inhibit HIF1α-controlled gene expression.

*Figure 5 continued on next page*

*Figure 5 continued*

**Figure supplement 1—source data 1.** Source data for *Figure 5—figure supplement 1B,C,D*.

## Discussion

Numerous studies have highlighted crucial roles of Hippo signaling in organ growth and size control, which also includes the functional characterization of this pathway in the growing vasculature. It was already shown that the loss of Yap1/Taz in ECs impairs endothelial proliferation and sprouting in the embryo and postnatal retina, whereas genetic gain-of-function experiments lead to endothelial hypersprouting and vascular hyperplasia (*Kim et al., 2017*; *Neto et al., 2018*; *Sakabe et al., 2017*; *Wang et al., 2017*). In ECs of the lymphatic system, Taz has been linked to maladaptive effects such as the loss of quiescence, aberrant entry into the cell cycle and, ultimately, cell death in response to disturbed flow (*Sabine et al., 2015*). Likewise, while protective laminar flow suppresses Yap1/Taz activity in cultured ECs, exposure to disturbed flow led to the induction of Yap1/Taz-dependent proliferative and proinflammatory responses (*Wang et al., 2016*). In the zebrafish embryo, blood flow was also identified as one of the upstream factors controlling the nuclear import of Yap1 in a Hippo-independent fashion (*Nakajima et al., 2017*). Another study has proposed that endothelial Yap1/Taz activity controls the expression of bone morphogenetic protein 4 (BMP4) and thereby transiently promotes intramembraneous ossification in the head of zebrafish embryos, while angiogenesis was unaffected (*Uemura et al., 2016*). These reports already suggest that the functional roles of Yap1/Taz in the vascular system are diverse and context-dependent. It is, nevertheless, striking that the EC-specific inactivation of the two genes led to opposite outcomes in the retina and in the skeletal system. This is reminiscent of the role of endothelial Notch signaling, which is a potent suppressor of angiogenesis in retina, brain and in tumors (*Benedito et al., 2012*; *Ridgway et al., 2006*) but promotes type H formation, vessel growth and osteogenesis in long bone (*Ramasamy et al., 2014*). While the exact cause of these disparate functional roles of the two pathways requires further investigation, local differences in blood flow and tissue oxygenation are likely to be relevant. Artery caliber, flow rates and calculated endothelial shear stress are comparably small inside bones (*Ramasamy et al., 2016b*) relative to other tissues so that the impact of hemodynamic parameters might be more limited in bone ECs. Nevertheless, the observed nuclear localization of Yap1 in the sinusoidal endothelium, which has the lowest rates of flow (*Bixel et al., 2017*; *Ramasamy et al., 2016b*), is still consistent with the previously reported flow-controlled regulation of Yap1/Taz.

The levels of tissue oxygenation also vary largely between different organs. While hypoxia is undetectable in the healthy retina and brain (*Kisler et al., 2017*; *Mezu-Ndubuisi et al., 2013*), the interior of bone is highly hypoxic. Live imaging in the adult murine calvarium showed the lowest local oxygen tension around the sinusoidal vasculature, whereas the endosteum is less hypoxic due to the presence of small arterioles (*Spencer et al., 2014*). In postnatal long bone, we observed that arterioles terminate in type H capillaries of the metaphysis and endosteum but not in sinusoidal (type L) vessels. Accordingly, multiple markers of hypoxia were absent from the metaphysis but highly abundant throughout the diaphysis (*Kusumbe et al., 2014*). Despite of these differences, we also found that the HIF pathway plays crucial roles in bone angiogenesis and the specification of type H ECs (*Kusumbe et al., 2014*; *Kusumbe et al., 2016*). The current study offers important clues that might help to explain the previous findings. Type H ECs in the metaphysis show low steady state levels of Yap1/Taz consistent with the rapid, Hippo-dependent degradation of these proteins. This might allow the necessary level of endothelial HIF signaling required for local EC proliferation and vessel growth in the metaphysis. In contrast, higher levels of nuclear Yap1/Taz in the ECs of the diaphyseal vasculature might suppress excessive activation of the HIF pathway even though the local microenvironment is highly hypoxic. This model is consistent with the observed upregulation of HIF-controlled genes in *Yap1/Taz$^{i\Delta EC}$* bone ECs and can also explain why the same mechanism does not apply to highly oxygenated tissues, such as retina, where HIF1$\alpha$ is unstable and rapidly degraded. It was proposed that Yap1 physically interacts with HIF1$\alpha$ with in cultured cancer cell lines (*Zhang et al., 2018*), whereas HIF2$\alpha$ was shown was shown to enhance Yap1 activity without direct interaction of the two proteins (*Jia et al., 2019*). Furthermore, it was shown that hypoxia increases the stability of Yap1 by inducing the degradation of Lats2 through the E3 ubiquitin ligase SIAH2 (*Ma et al., 2015*). The same study also proposed that HIF1$\alpha$ and Yap1 can interact directly, leading to stabilization of

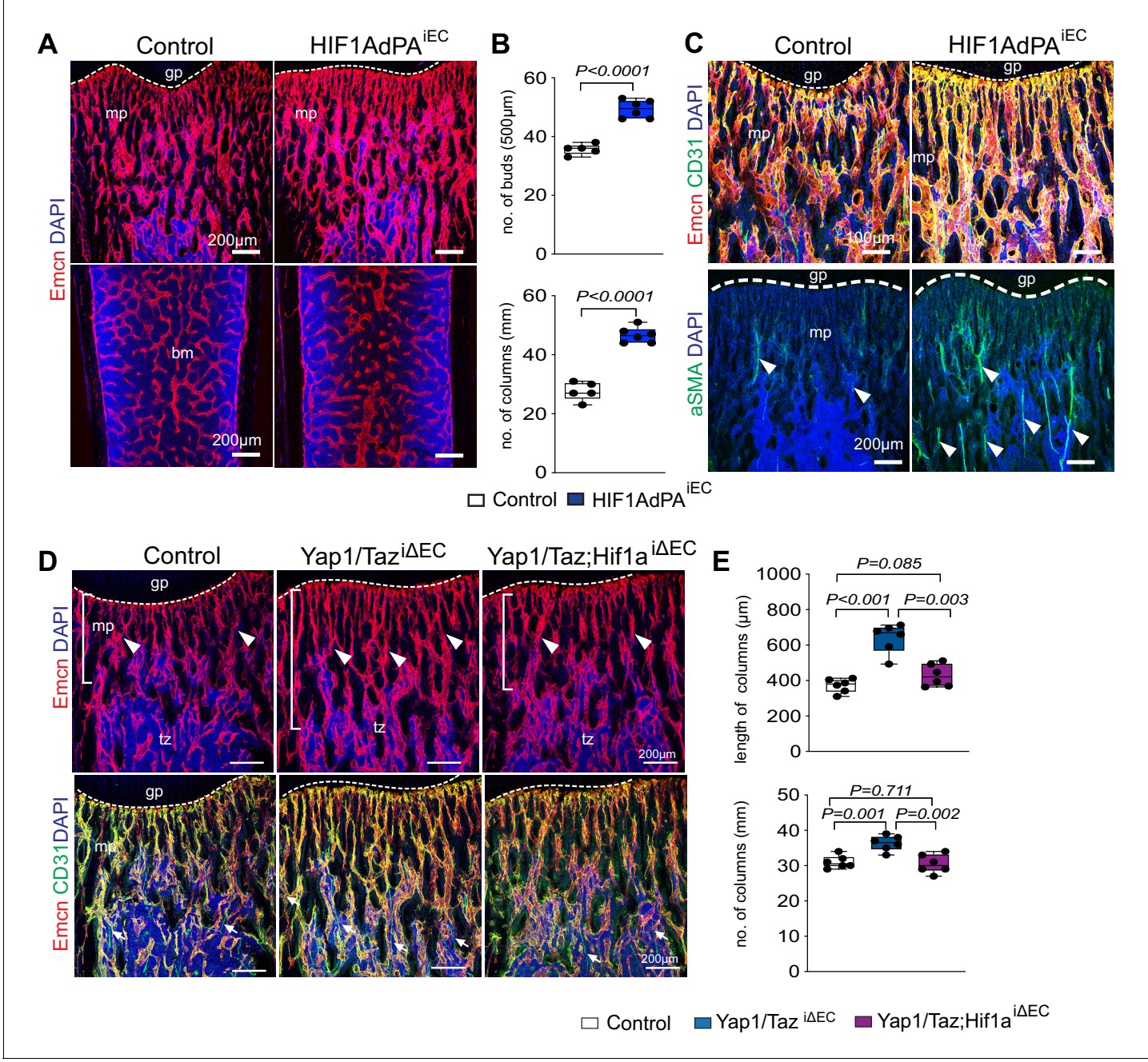

**Figure 6.** Cross-talk between Yap1/Taz and HIF1α in bone ECs. (**A, B**) Representative confocal images of control and HIF1AdPA^iEC femur. Emcn+ (red) ECs and nuclei (DAPI, blue) are stained (**A**). Vessel buds and column are increased in the HIF1AdPA^iEC metaphysis relative to littermate control (**B**) (n = 5; data are presented as mean ±sem, *P* values, two-tailed unpaired *t-test*). (**C**) Metaphyseal Emcn^hi (red) CD31^hi (green) capillaries in P21 control and *HIF1A* gain-of-function (HIF1AdPA^iEC) femur. Nuclei, DAPI (blue). Number of aSMA positive arteries is increased in HIF1AdPA^iEC femur compared to control. (**D, E**) Maximum intensity projections of Yap1/Taz^iΔEC, Yap1/Taz; Hif1a^iΔEC and control femur stained for Emcn (red) and CD31 (green). Nuclei, DAPI (blue). Arrowheads mark vessels in transition zone (**D**). Quantitative analysis of length and number of vessel column (**E**) (n = 6; data are presented as mean ±sem, *P* values, two-tailed unpaired *t-test*).

The online version of this article includes the following source data for figure 6:

**Source data 1.** Source data for *Figure 6B,E*.

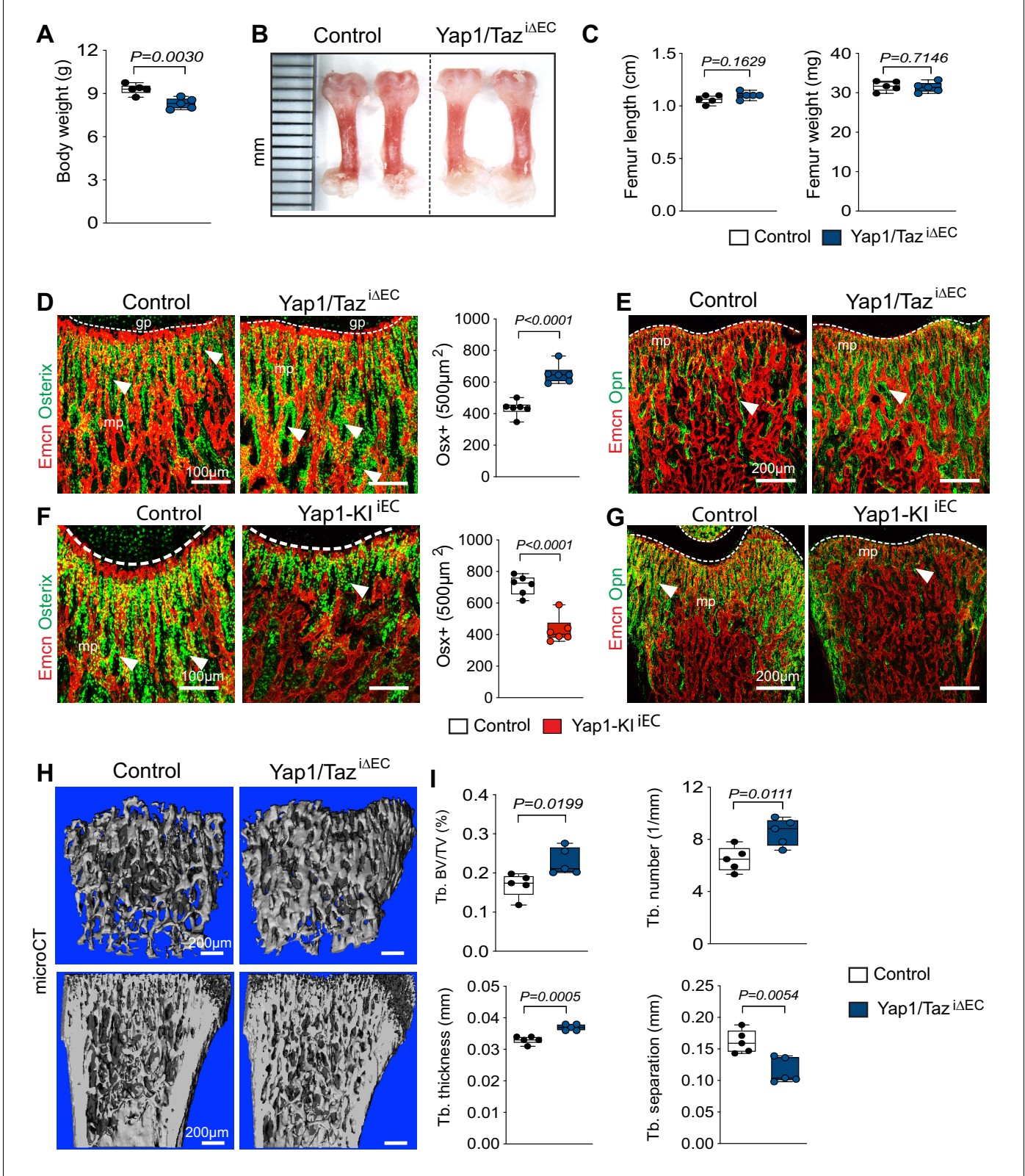

**Figure 7.** Endothelial Hippo signaling promotes coupling of angiogenesis and osteogenesis. (**A**) Average body weight of P21 control and Yap1/Taz$^{i\Delta EC}$ mutants (n = 6; data are presented as mean ±sem, *P* values, two-tailed unpaired *t-test*). (**B, C**) Representative images of P21 control and Yap1/Taz$^{i\Delta EC}$ femur (**C**). Quantitation of femur length and weight show no changes between Yap1/Taz$^{i\Delta EC}$ mutants and littermate controls (**C**) (n = 6; data are presented as mean ±sem, *P* values, two-tailed unpaired *t-test*). (**D**) Confocal images showing Osterix+ cells (green) in relation to Emcn+ ECs (red) in P21

*Figure 7 continued on next page*

*Figure 7 continued*

Yap1/Taz$^{i\Delta EC}$ and control metaphysis. Graph on the right shows quantitative analysis of Osterix+ (Osx+) cells (control n = 6 and Yap1/Taz$^{i\Delta EC}$n = 6; data are presented as mean ±sem, *P* values, two-tailed unpaired *t-test*). (E) Confocal images showing bone vessels (Emcn), and the bone matrix protein Osteopontin in Yap1/Taz$^{i\Delta EC}$ vs control femur. (F) Representative confocal images of Osterix+ cells (green) in relation to Emcn+ ECs (red) in the P21 control and Yap1-KI$^{iEC}$ femoral metaphysis. Graph on the right shows significant reduction of Osterix+ cells in Yap1-KI$^{iEC}$ mutants (n = 6; data are presented as mean ±sem, *P* values, two-tailed unpaired *t-test*). (G) Decreased bone matrix protein Osteopontin (Opn, green) deposition in P21 Yap1-KI$^{iEC}$ femur relative to control. ECs, Emcn (red). (H, I) Representative µCT images of trabecular bone in P21 control and Yap1/Taz$^{i\Delta EC}$ femur (H). Quantitative analysis of trabecular volume (BV/TV, bone volume/total volume) trabecular (Tb.) number, Tb. thickness, and Tb. separation (I). (n = 5; data are presented as mean ±sem, *P* values, two-tailed unpaired *t-test*).

The online version of this article includes the following source data and figure supplement(s) for figure 7:

**Source data 1.** Source data for *Figure 7A,C,D,F,I*.
**Figure supplement 1.** Endothelial Yap1/Taz controls coupling of angiogenesis and osteogenesis.
**Figure supplement 1—source data 1.** Source data for *Figure 7—figure supplement 1A,C,E,F*.

HIF1α in cells cultured under hypoxic conditions. While these findings would not explain the observed repression of HIF1α activity in bone ECs, it was also reported that Yap1/Taz can act as transcriptional co-repressors and inhibit the expression of hypoxia-induced genes (*Kim et al., 2015*). While the exact mechanism requires further investigation, it is increasingly evident that the activity of the two pathways is linked und therefore influenced by differences in oxygenation in healthy organs but, potentially, also in response to pathological processes involving tissue ischemia.

Apart from showing that Yap1/Taz can act as negative regulators of growth processes in the bone, our study highlights the stringent coupling of angiogenesis and osteogenesis in the skeletal system. Our findings also raise the interesting question whether drugs acting on the Hippo pathway, such as Verteporfin, which is a small molecule inhibitor of Yap1–TEAD complex formation and of Yap1/Taz-mediated cell proliferation (*Kimura et al., 2016*; *Liu-Chittenden et al., 2012*), might be therapeutically useful to increase bone mineral density. This could be relevant in the context of aging-related loss of mineralized bone or in osteoporosis, which leads to bone weakness, increased risk of fracturing, loss of mobility and chronic pain. Future studies will have to explore this important topic, which is likely to require the development of targeted therapeutic strategies to avoid adverse effects in the many different cell types and organs utilizing the Hippo pathway.

## Materials and methods

The Key Resources Table (*Supplementary file 1*) provides a list with the mouse strains, cell line, antibodies, reagents, kits and software used for this study.

### Animal models

C57BL/6J male mice were used for all wild-type bone analysis. For pharmacological treatments, both male and female mice bone were analysed. For the inducible and EC-specific inactivation of the *Yap1* and *Taz/Wwtr1* in mice, we bred *Yap1*$^{lox/lox}$ and/or *Taz*$^{lox/lox}$ conditional mutants (*Reginensi et al., 2013*) to *Cdh5-CreERT2* transgenic mice (*Kusumbe et al., 2015*; *Langen et al., 2017*) to generate Cre-positive Yap1/Taz$^{i\Delta EC}$ double mutants. Mice carrying loxP-flanked *Lats2* alleles (*Lu et al., 2010*) were also bred to the *Cdh5-CreERT2* line to generate EC-specific Lats2$^{i\Delta EC}$ mutants. To express stabilized Hif1α in ECs (HIF1AdPA$^{iEC}$), we interbred *R26HIF1AdPA* (*Kim et al., 2006*) and *Cdh5-CreERT2* mice. Using *Hif1a*$^{lox/lox}$ mice (*Ryan et al., 2000*) with a similar breeding strategy, we generated inducible and EC-specific *Yap/Taz Hif1a* triple loss-of-function mutants (Yap1/Taz Hif1a$^{i\Delta EC}$). In all these experiments, Cre-negative littermates were used as controls.

To generate EC-specific *Cdh5-mT/nG* transgenic reporter mice expressing membrane-anchored tomato protein and nuclear green fluorescent protein, a cassette consisting of membrane-tagged tdTomato (Addgene plasmid #17787), 2A peptide, AU1 tag and H2B-EGFP (Addgene plasmid #11680) followed by a polyadenylation signal sequence and a FRT-flanked ampicillin resistance cassette were introduced by recombineering into the start codon of a large *Cdh5* genomic fragment in PAC clone 353-G15. After Flp-mediated excision of the ampicillin resistance cassette in bacteria, positive clones were validated by PCR analysis and used in circular form for pronuclear injection into fertilized mouse oocytes. Founders were screened by PCR and immunostaining with Endomucin

(Emcn) in bone. Genotypes of mice were determined by PCR. *Cdh5-mT/nG* transgenic reporter mice were introduced into the Yap1/Taz$^{i\Delta EC}$ double and Lats2$^{i\Delta EC}$ mutant background and the corresponding controls.

To generate constitutive-active Yap knock-in (Yap-KI) mice, we mutated Yap1 amino acid 112 serine to alanine (Yap1$^{S112A}$), as this phosphorylation site is responsible for the translocation of Yap1 from nucleus to cytosol. The *Yap1$^{S112A}$* cDNA was inserted into a CAG-STOP-eGFP-ROSA26TV (Addgene plasmid #15912) vector and recombined into bacterial artificial chromosome (BAC) clone containing the murine Rosa26 locus. Linear recombined clones were injected into fertilized mouse oocytes. Founders were screened by PCR and EC-specific Yap1 nuclear localization was conformed in retina.

All animal experiments were performed according to the institutional guidelines and laws, approved by local animal ethical committee and were conducted at the University of Münster and the Max Planck Institute for Molecular Biomedicine with permissions (84–02.04.2015.A185, 84–02.04.2016.A160, 81–02.04.2017.A238) granted by the Landesamt für Natur, Umwelt und Verbraucherschutz (LANUV) of North Rhine-Westphalia. Animals were combined in groups for experiments irrespective of their sex.

## Tamoxifen-inducible Cre-mediated recombination

Pups received daily intraperitoneal injections (IP) of 50 μg or 100 μg of tamoxifen (*Pitulescu et al., 2010*) from postnatal day 1 (P1) to P3 or from P6 to P8. Tamoxifen (Sigma, Cat#T5648) stocks were prepared by dissolving 20 mg in 500 μl of ethanol and vortexing for 10mins before an equal volume of Kolliphor EL (Sigma, Cat#C5135) was added. 1 mg aliquots were stored at −20°C and dissolved in the required volume of PBS prior to injection.

## Bone sample preparation

Mice were sacrificed and long bones (femur and tibia) were harvested and fixed immediately in ice-cold 2% paraformaldehyde (PFA) for 6 to 8 hr under gentle agitation. Bones were decalcified in 0.5M EDTA for 16 to 24 hr at 4°C under agitation, which was followed by overnight incubation in sucrose solution (20% sucrose, 2%PVP) and mounted in bone mounting medium (8% galatine, 2% PVP). Samples were stored overnight at −80°C. 60–100 μm-thick cryosections were prepared for immunofluorescence staining (*Kusumbe et al., 2015*).

## Pharmacological treatments

Three-week-old wild type mice were injected intraperitoneally either with 50 μg/g MG132 (Millipore, Cat# 474790) or DMSO only (vehicle control). Three hours after injection, mice were sacrificed and femurs were dissected and processed for immunostaining as described below. For labeling of hypoxic cells, mice were intraperitoneally injected with 60 mg/kg Pimonidazole (Hypoxyprobe Inc) 2 hr before analysis.

## Proliferation assay in vivo

For the analysis of proliferating cells in bone in vivo, mice received an intraperitoneal injection of 300 μg of EdU for 3 hr before analysis. For retina, P6 pups received 100 μg of EdU for 2 hr. Detection of proliferating cells in fixed bone sections and whole mount retina was achieved by staining with Click-iT-EdU Alexa-647 imaging kit (Invitrogen, Cat# C10340) according to the manufacturer's instructions.

## HUVEC culture and siRNA mediated transfection

Human umbilical vein endothelial cells (HUVEC) (ThermoFisher Scientific, Cat# C0035C), certified by the supplier to be free of mycoplasma and pathogens, were cultured in EGM2 complete medium (Lonza, Cat# CC-3156) along with growth factor supplements (Lonza, Cat# CC-4176) at 37°C in 5% $CO_2$ in a humidified atmosphere. Cells at less than 4–5 passages were used for all experiments. For siRNA-mediated knockdown, HUVECs were transfected using the Lipofectamine RNAiMAX (Invitrogen, Cat# 13778075) with target gene-specific siRNA target sequences or control siRNA (Negative Control #1 siRNA (Cat#AM4611); *YAP1* #1: (5′−3′) GGUGAUACUAUCAACCAAAtt (ID: s20366); *YAP1* #2: (5′−3′) ACAGUCUUCUUUUGAGAUAtt (ID:s20367); *TAZ* (*WWTR1*) #1: (5′−3′) G

UACUUCCUCAAUCACAUAtt (ID:s24789); *TAZ* (*WWTR1*) #2: (5'−3') GGAUACAGGAGAAAACGCA tt (ID:s24787); *HIF1A* #1: (5'−3') CCAUAUAGAGAUACUCAAAtt (ID:s6539) from ThermoFisher Scientific. For hypoxia experiments, siRNA-transfected cells were grown in 1% $O_2$ and 5% $CO_2$ at 37℃ for 24 hr in a humidified atmosphere.

## Immunofluorescence staining

For immunostaining, slides with bone sections or sections from other organs were washed in ice-cold PBS and permeabilized with ice-cold 0.3% Triton-X-100 in water for 10 mins at room temperature (RT). Samples were incubated in blocking solution (5% heat-inactivated donkey serum in 0.3% Triton-X-100) for 30mins at RT. Primary antibodies (rat monoclonal anti-Endomucin (V.7C7) (Santa Cruz, Cat# sc-65495), goat polyclonal anti-CD31 (R and D, Cat# AF3628), rabbit monoclonal anti-Yap1 (Cell Signaling, Cat# 14074), rabbit polyclonal anti-Wwtr1 (Sigma-Aldrich, Cat# HPA007415), rabbit monoclonal anti-Yap(ser127) (Cell Signaling, Cat# 13008), rabbit monoclonal anti-Yap1/Taz (D24E4) (Cell Signaling, Cat# 8418), rabbit polyclonal anti-Lats2 (Bethyl, Cat#A300-479A), rabbit polyclonal anti-Hif-1α (Thermo Scientific, Cat# Pa1-16601), rabbit polyclonal anti-Hif-2α (Novus Biologicals, Cat# NB100-122), mouse monoclonal anti-alpha-smooth muscle actin (Cy3- conjugated; Sigma-Aldrich, Cat#C6198), goat polyclonal anti-VEGFR3 (R and D, Cat# AF743), Rabbit polyclonal anti-Osterix/sp7 (Abcam, Cat# ab22552), Goat polyclonal anti-Osteopontin (R and D, Cat# AF808), Rabbit polyclonal anti-Osteocalcin (Lifespan Bioscience, Cat# LS-C17044), Rabbit polyclonal anti-Runx2 (M-70) (Santa Cruz, Cat# sc-10758), Rabbit monoclonal anti-ATP6VIB1 and 2 (Abcam, Cat#Ab200839) were diluted in 5% donkey serum and PBS and incubated overnight at 4℃. Next, slides were washed 3 to 5 times in PBS in 5–10 min intervals. Species-specific Alexa Fluor secondary antibodies (ThermoFisher) diluted in PBS were added and incubated for 3 hr at RT.

For whole-mount retina staining, P6 eyes were removed and fixed in ice cold 4% PFA for 2 hr. After fixation, retinas were dissected and processed as described previously (*Pitulescu et al., 2010*). After two washes with ice-cold PBS, samples were incubated in blocking buffer (1%BSA, 1% Triton X-100, 3% heat inactivated donkey serum in PBS) for 1 hr on rotating shaker. Next, blocking buffer was replaced by Pblec buffer (1 mM $CaCl_2$, 1 mM $MgCl_2$, 0.1 mM $MnCl_2$, 0.1% Triton X-100 in PBS). Isolectin-B4 (IB4; Vector, Cat#Ab200839) or rabbit monoclonal anti-ERG (Abcam, Cat# ab22552) were diluted in Pblec buffer and each retina was incubated in 100 µl of solution overnight at 4℃. Next, samples were washed five times in incubation buffer (diluted blocking buffer 1:1 in PBS) and incubated with the appropriate Alexa Fluor488, 546 and 594-conjugated secondary antibodies for 2 hr at RT. Later, retinas were washed five times with ice-cold PBS and mounted under a stereomicroscope.

## Fluorescence-activated cell sorting (FACS)

Single cell suspensions were prepared from femur and tibia as described (*Langen et al., 2017*). CD31-APC (Goat polyclonal anti-CD31 (APC-conjugated; R and D, Cat# AFB3628A) coupled and Emcn primary antibodies were added to the single cell suspension and incubated for 45mins on ice. Samples were washed two to three times with blocking solution. Secondary anti rat-PE antibody and DAPI (Sigma, Cat#D9542) were added and the incubation continued for 45mins on ice. Next, samples were washed 2–3 times with blocking solution and resuspended using ice-cold PBS. Cell sorting was performed on a FACS Aria II cell sorter (BD Bioscience). Dead cells and debris were excluded by FSC, SSC and DAPI positive signal. Sorted bone ECs were collected in RTL buffer for RNA isolation. FACS data were analysed with FlowJo Software (FLOWJO, LLC).

## Quantitative PCR

Total RNA was isolated from HUVEC or sorted bone ECs using RNA Plus mini kit (Qiagen, Cat#74134) according to the manufacturer's instructions. RNA was reverse-transcribed using the iScript cDNA synthesis kit (Bio-Rad, Cat#1708890). Quantitative PCR was carried out using gene TaqMan Gene Expression Master Mix (ThermoFisher Scientific, Cat#4369016) and specific Taqman probes human: eukaryotic 18S rRNA (4319413E), VEGFA (Hs00900055_m1, ANGPTL4 (Hs01101127_m1), IGFBP2 (Hs01040719_m1), XBP1 (Hs00231936_m1), CTGF (Hs01026927_g1), CYR61(Hs00998500_g1), YAP1(Hs00902712_g1), WWTR1(Hs00210007_m1), HIF1A(Hs00153153_m1) and mouse probes: Vegfa (Mm00437306_m1), Angptl4 (Mm00480431_m1), Ctgf (Mm01192932_g1),

Cyr61 (Mm00487499_g1) from ThermoFisher Scientific (Cat# 4331182) using a C1000 Touch Thermal cycler (BIORAD).

Primer sequences for qPCR analysis of *Yap1* and *Wwtr*1 expression in freshly isolated bone ECs (*Figure 2—figure supplement 1B*) are provided in the Key Resources Table (*Supplementary file 1*).

## Immunoprecipitations and immunoblotting

For immunoprecipitations, HUVEC cells were lysed in radioimmunoprecipitation (RIPA) lysis buffer (150 mM NaCl, 25 mM Tris-HCl, 1 ml of 0.5 M EDTA pH 8.0, sodium dodecyl sulfate (SDS), 1% sodium deoxycholate and 0.25% Triton X-100) containing protease inhibitor (cOmplete, Roche) and phosphatase inhibitor tablets (PhosSTOP, Roche). Lysates were centrifuged at 4°C for 30 min at 20,000 g, and aliquots were set aside for direct input blot analysis. For the immunoprecipitation, the remaining lysates were pre-cleared for 1 hr at 4°C with Dynabeads protein-G. Pre-cleared lysates were incubated with primary antibody for 2 hr at 4°C, after which the beads were added and incubated for an additional 2 hr at 4°C. Immunocomplexes were washed five times with lysis buffer (without SDS and deoxycholate) and analysed by SDS–PAGE.

For immunoblotting, cell lysates were boiled in Laemmli buffer for 5mins and then used for immunoblotting. Proteins were separated in 6 or 8% SDS – PAGE, then transferred onto nitrocellulose membranes, followed by blocking for 1 hr and overnight incubation with primary antibodies. After washing, membranes were incubated with horseradish peroxidase-conjugated secondary antibodies for 1 hr. Antibody binding was visualized by enhanced chemiluminescence reagent (Millipore) using Fuji medical X-ray films.

## Chromatin immunoprecipitation (ChiP)

Chromatin immunoprecipitation was performed as described previously (*Sivaraj et al., 2013*). In brief, siRNA siControl and *siYAP1/WWTR1* transfected HUVEC were grown in 1% $O_2$ for 24 hr, then cross-linked with 1% formaldehyde for 10 min at room temperature. Samples were then sonicated into 200–700 bp fragments using a Branson digital sonifier and chromatin was immunoprecipitated with 5 μg of anti-rabbit IgG or anti-HIF1α, followed by reverse cross-linking. The recovered DNA was purified using the Qiagen DNA isolation kit and DNA was analyzed using qPCR. The *VEGFA* promoter HRE sequence was amplified using 5'-GCCAGACTCCACAGTGCATA-3'and 5'-CTGA-GAACGGGAAGCTGTGT-3' primer pair. The DNA recovered from chromatin that was not immunoprecipitated was used as input.

## RNA sequencing and data analysis

Three-week-old bone ECs were sorted from *Cdh5-mTnG* metaphysis (mpECs) and diaphysis (bmECs) or Yap1/Taz$^{i\Delta EC}$ and Lats2$^{i\Delta EC}$ mutant bone with their respective littermate controls. RNA was isolated using the RNeasy Plus Micro Kit (QIAGEN, Cat# 74134) according to the manufacturer's instructions. RNA quality was checked using a 2100 BioAnalyzer (Agilent). 100 ng of RNA were used for preparation of sequencing libraries with the TruSeq Stranded Total RNA Library Prep Kit (Illumina) according to the manufacturer's instructions. Libraries were validated using a BioAnalyzer, quantified by qPCR and Qubit Fluorometric Quantitation (Thermo Fisher Scientific, Cat#Q32851). Libraries were diluted to a final concentration of 15pM for sequencing. The MiSeq Reagent Kit v3 (Illumina, Cat#MS-102–3001) was used for sequencing with a MiSeq (Illumina). Biological triplicates were used.

RNA-seq data analysis was performed as described previously (*Langen et al., 2017*) with some modifications. The quality assessment of raw sequence data was performed using FastQC (Version: FastQC 0.11.3). Paired-end sequence reads were mapped to the mm10 mouse genome assembly (GRCm38) using TopHat-2 (Version: tophat-2.0.13). The mouse genome was downloaded from the iGenome portal. HTSeq was used to count the aligned reads on a per gene basis (Version: HTSeq-0.6.1).

The count data were normalized using the Variance Stabilizing Transformation (VST) function from the DESeq2 package. Principal Component Analysis (PCA) was performed on transformed read counts using the variable genes to assess the overall similarity between the samples. Differential gene expression analysis between control and mutants were performed using DESeq2. Differentially expressed genes were selected using a FRD-adjusted p-value cut-off <0.01 and an absolute $\log_2$

fold change >0.5. Gene symbols were annotated using biomart (BioConductor version 3.1). Gene ontology analysis and cellular signaling pathways were performed with the Enrichr online tool (http://amp.pharm.mssm.edu/Enrichr/). Heat maps were generated with http://heatmapper.ca/.

RNA-sequencing data of control and mutant bone ECs were uploaded to the Gene Expression Omnibus (GEO) under the accession number GSE102181.

## Quantification and statistical analysis

Immunostained bone sections were imaged with a Leica SP8 confocal microscope and the following settings: $1024 \times 1024$; 200 speed, low magnification image z stack- 4 µm; high magnification stack: 2 µm. Images were analysed, quantified and processed using Volocity (Perkin Elmer), Adobe CS6 Photoshop and Illustrator software.

For quantification of vascular alterations in bone, the number of vessel buds or columns and EC proliferation were quantified manually using Volocity software (Perkin Elmer). Quantitation is based on 3–5 images of bone sections per animal and average values per animals were combined in the graphs.

Statistical analysis was performed using GraphPad Prism software or the R statistical environment (http://r-project.org). All data are presented as mean ± s.e.m. unless indicated otherwise. Unpaired two tailed student t-tests were used to determine statistical significance. $p < 0.05$ was considered significant unless stated otherwise. Sample number was chosen based on experience from previous experiments. Reproducibility was ensured by several independent experiments. No animals were excluded from analysis.

## Acknowledgements

We thank J Wrana for floxed *Yap1* and *Wwtr1/Taz* mutant mice, J Platzek for technical assistance, and M Stehling for EC sorting. Funding was provided by the Max Planck Society, the University of Münster and the European Research Council (AdG 339409 AngioBone; AdG 786672 PROVEC).

## Additional information

### Competing interests

Gou Young Koh: Reviewing editor, *eLife*. The other authors declare that no competing interests exist.

### Funding

| Funder | Grant reference number | Author |
|---|---|---|
| H2020 European Research Council | AdG 786672 PROVEC | Ralf H Adams |
| Max-Planck-Gesellschaft | Open-access funding | Ralf H Adams |

The funders had no role in study design, data collection and interpretation, or the decision to submit the work for publication.

### Author contributions

Kishor K Sivaraj, Conceptualization, Formal analysis, Investigation, Methodology, Writing - original draft, Writing - review and editing; Backialakshmi Dharmalingam, Vishal Mohanakrishnan, Silke Schröder, Investigation, Methodology; Hyun-Woo Jeong, Data curation, Formal analysis, Investigation, Methodology; Katsuhiro Kato, Formal analysis, Investigation; Susanne Adams, Resources, Methodology; Gou Young Koh, Conceptualization, Resources; Ralf H Adams, Conceptualization, Resources, Supervision, Funding acquisition, Writing - original draft, Project administration, Writing - review and editing

Author ORCIDs
Kishor K Sivaraj (iD) https://orcid.org/0000-0002-5321-3400
Ralf H Adams (iD) https://orcid.org/0000-0003-3031-7677

Ethics

Animal experimentation: All animal experiments were performed according to the institutional guidelines and laws, approved by local animal ethical committee and were conducted at the University of Münster and the Max Planck Institute for Molecular Biomedicine with permissions (84-02.04.2015.A185, 84-02.04.2016.A160, 81-02.04.2017.A238) granted by the Landesamt für Natur, Umwelt und Verbraucherschutz (LANUV) of North Rhine-Westphalia. All procedures performed with great care and every effort was made to minimize suffering.

Decision letter and Author response
Decision letter https://doi.org/10.7554/eLife.50770.sa1
Author response https://doi.org/10.7554/eLife.50770.sa2

## Additional files

### Supplementary files
• Supplementary file 1. Key Resources Table.
• Transparent reporting form

### Data availability
RNA-sequencing data were uploaded to Gene Expression Omnibus (GEO) database. The accession number GSE102181 and can be accessed at: https://www.ncbi.nlm.nih.gov/geo/query/acc.cgi?acc=GSE102181.

The following datasets were generated:

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
