## [Decision Letter]

**Acceptance summary:**

We believe that your study delineating mechanisms of cross-talk between YAP1/TAZ and HIF1α in the control of bone angiogenesis is well done and elegant. It is provides novel insights, in particular, a meaningful molecular mechanism to the interface between osteogenesis and angiogenesis. We therefore believe that your study will likely attract the attention of both bone and vascular biologists and contribute, more generally, to the expanding domain of integrative physiology. We were particularly impressed with the thoughtful and thorough responses to the review critique, and find no remaining issues.

**Decision letter after peer review:**

Thank you for submitting your article "YAP1 and TAZ negatively control bone angiogenesis by limiting hypoxia-inducible factor signalling in endothelial cells" for consideration by *eLife*. Your article has been reviewed by three peer reviewers, and the evaluation has been overseen by Mone Zaidi as Reviewing Editor and Clifford Rosen as the Senior Editor. The following individual involved in review of your submission has agreed to reveal their identity: Geert Carmeliet (Reviewer #3).

Below are specific comments relating to the decision to help you prepare a revised submission.

Summary:

Whereas all three reviewers felt that the manuscript provided new and insightful information relating to the role of YAP1 and TAZ in bone vascularization, and was well written with carefully conducted experiments, all had concerns that merit careful revaluation.

Essential revisions:

1) A overall major concern for reviewers 2 and 3 relates to similar observed effects of deleting YAP1, TAZ or Lats2 in both the metaphysis and diaphysis, particularly as the diaphysis is more hypoxic than the metaphysis with a higher expression of hypoxia target gene in endothelial cells. The request from both reviewers is for you to carefully analyze blood vessels and sorted cells from the respective regions in the mutants, including the YAP knock-in mice. Reviewer 3 further suggests that you examine YAP1/TAZ and Hippo target gene expression in sorted endothelial cells from HIF1α conditional mutants.

2) Reviewer 2 indicates (and I fully agree) that co-immunoprecipitation of YAP1/TAZ and HIF1α is not enough to establish physical binding that leads to HIF1α inhibition. Establishing physical binding, short of an X-ray crystal, would require a biophysical method, such as a protein thermal shift assay with recombinant proteins in solution. Furthermore, it is suggested that you determine other molecular partners of a potential complex, including ARNT and HIF2α.

3) Reviewer 2 suggests that you expand the analysis of gene expression to other HIF target genes beyond *VEGFA*.

4) Reviewer 3 suggests that you further analyze YAP1/TAZ signaling and angiogenesis in mutants with inactivated HIF1α.

5) Reviewer 3 requests further data on bone parameters, considering your earlier elegant work on angiogenesis-osteogenesis coupling. We will understand if you do not wish to provide this data and focus the manuscript to vascularization – nonetheless, it is felt that this aspect will considerably enhance the value of your findings to a more general audience.

---

## [Author Response]

Essential revisions:1) A overall major concern for reviewers 2 and 3 relates to similar observed effects of deleting YAP1, TAZ or Lats2 in both the metaphysis and diaphysis, particularly as the diaphysis is more hypoxic than the metaphysis with a higher expression of hypoxia target gene in endothelial cells. The request from both reviewers is for you to carefully analyze blood vessels and sorted cells from the respective regions in the mutants, including the YAP knock-in mice. Reviewer 3 further suggests that you examine YAP1/TAZ and Hippo target gene expression in sorted endothelial cells from HIF1α conditional mutants.

We would like to thank the reviewers for this excellent suggestion and agree that it is very interesting to explore the differences between metaphysis (mp) vs. diaphysis (dp) and different EC subpopulations. We have added new data showing the expression of Hippo and HIF target genes in freshly sorted type H and type L ECs from different mutant models. Consistent with the nuclear 2 localization of Yap1/Taz in dp ECs (shown in Figure 1J) and the rapid (*Lats2*-controlled) degradation of the transcriptional co-regulators in mp ECs (shown in Figure 1I), expression the Hippo targets *Ctgf* and *Cyr61* is much higher in type L than in type H ECs (see new data in Figure 4H, I). Loss of Yap1/Taz strongly reduces *Ctgf* and *Cyr61* in both EC subsets, whereas the inactivation of *Lats2* results in increased expression of the two genes. Consistent with the effect of hypoxia of Yap1/Taz nuclear localization and Hippo target gene expression seen in vitro (Figure 5A, B), this effect was strongest in type L ECs (Figure 4I). Furthermore, EC-specific inactivation of *Hif1a* reduces both *Ctgf* and *Cyr61* expression in type L ECs (Figure 5—figure supplement 1; please note that only the *Ctgf* result is statistically significant) in addition to the expected loss of HIF target gene expression (*Vegfa* and *Anptl4*). Consistent with our other data, loss of Yap1/Taz increases endothelial *Vegfa* and *Anptl4* expression, which is, again, most obvious for type L ECs coming from the highly hypoxic bone marrow cavity. Loss of *Lats2* strongly diminishes *Vegfa* and *Anptl4* expression. We have not been able to investigate Yap1^S112A^ overexpressing mice within the allotted time window because we simply did not have enough mice available for the experiment. However, as the phenotype of these gain-of-function mutants is similar but milder than that the *Lats2* loss-of function model, we do not expect any novel insights from this particular experiment.

2) Reviewer 2 indicates (and I fully agree) that co-immunoprecipitation of YAP1/TAZ and HIF1α is not enough to establish physical binding that leads to HIF1α inhibition. Establishing physical binding, short of an X-ray crystal, would require a biophysical method, such as a protein thermal shift assay with recombinant proteins in solution. Furthermore, it is suggested that you determine other molecular partners of a potential complex, including ARNT and HIF2α.

We agree that our data does not establish that there is a direct physical interaction between Yap1 (or Taz) and HIF1α. Instead, it is entirely feasible that the link is not direct and involves other proteins. As suggested, we have attempted the thermal shift assay (see Author response image 1). This approach worked in our hands for control experiments but did not provide meaningful results with recombinant Yap1 and HIF1α. This finding, however, does not disprove the possibility of a physical interaction, which might require protein modification or additional factors that could be missing in our simple assay. We have also expanded our discussion of the existing literature on this subject (Discussion, second paragraph) to provide readers with sufficient background information

In addition, we have included a new piece of data, which supports that Yap1/Taz control HIF1α biological activity. Knockdown of YAP1/TAZ in HUVECs cultured under hypoxic conditions increases HIF1α binding to a hypoxia response element (HRE) in the *VEGFA* promoter (Figure 5F). While this result does not shed new light on the existence or absence of a physical interaction, it argues that the ability of HIF1α to bind DNA (or HREs, to be more precise) is modulated by Yap1/Taz.

3) Reviewer 2 suggests that you expand the analysis of gene expression to other HIF target genes beyond VEGFA.

Agree. We now show data on *ANPTL4, IGFBP2* and *XBP1* in addition to *VEGFA* (Figure 5E and Figure 5—figure supplement 1C).

4) Reviewer 3 suggests that you further analyze YAP1/TAZ signaling and angiogenesis in mutants with inactivated HIF1α.

Agree. We have expanded the analysis of these mutants, which also includes the analysis of Osterix+ cells (in connection with the next question). The new data are consistent with our original conclusion and show that inactivation of *Hif1a* leads to normalization of the exaggerated angiogenesis seen in Yap1/Taz mutant bone (Figure 6D, E and Figure 7—figure supplement 1F).

5) Reviewer 3 requests further data on bone parameters, considering your earlier elegant work on angiogenesis-osteogenesis coupling. We will understand if you do not wish to provide this data and focus the manuscript to vascularization – nonetheless, it is felt that this aspect will considerably enhance the value of your findings to a more general audience.

We are very happy to provide the requested data, which includes the staining of Osterix+ cells and µCT analysis of mineralized bone in Yap1/Taz mutants. In a nutshell, these data shown that enhanced angiogenesis in the loss-of-function mutants results in increased osteogenesis, whereas the number of Osterix+ cells and Osteopontin deposition are reduced in Yap1 gain-of-function animals (Figure 7D-I).